# Phrasal Recurrent Neural Network

### Abstract

We propose a new, simple, yet effective framework, phrasal recurrent neural networks (pRNN), for language modeling and machine translation. Different from previous RNN-based language models, pRNNs store the sentential history as a set of candidate phrases with different lengths that precede the word to predict. To represent phrases as fix-length real-valued vectors, we build the RNN pyramid, which is composed of shifted parallel RNN sequences. When predicting the next word, pRNNs employ a soft attention mechanism to selective and combine the suggestions of candidate phrases. We test our model on language model and machine translation tasks. Our model leads to an improvement of over 10 points in perplexity both on standard Penn Treebank and FBIS English data set over a state-of-the-art LSTM language modeling baseline. We also apply pRNNs to encode the source sentence of machine translation besides a conventional bi-direction encoder, which improves over the Moses (phrase-based statistical model) and a strong sequence-to-sequence baseline in the Chinese-English machine translation task.

## 1 Introduction

There are latent nest structures beyond sequential surface words in natural language (Chomsky, 1957). In the last two decades, researchers incorporated more and more rich structural information into conventional language model (Jelinek and Lafferty, 1991; Chelba, 1997; Chelba and Jelinek, 2000; Emami and Jelinek, 2005) and more recently on neural network-based language modeling (Dyer et al., 2016). Among the effort, one direction is to explore sub-word structures(Costa-Jussà and Fonollosa, 2016; Chung et al., 2016; Luong and Manning, 2016; Sennrich et al., 2015), such as characters, mostly to handle out-of-vocabulary word problem. A somewhat opposite direction is to explore hyper-word structure. For example, (Eriguchi et al., 2016) adopts parsing tree in encoder phase in machine translation, (Stahlberg et al., 2016) proposes to use hierarchical phrase-based (HPB) model to guide the search in decoding. Both models, however, rely heavily on human labeled data on the language structures, which is extremely expensive and limited in scale.

In this paper, we propose **phrasal recurrent neural networks** (pRNNs; §2), a general framework of RNNs (Elman, 1990) that explicitly models task-specific nested phrases from plain text. Here we use "phrase" as its definition in phrase-based statistical machine translation (PB-SMT(Zens et al., 2002; Koehn et al., 2003)), which indicates any continues sequences of words. What different here are pRNNs permit phrases with arbitrary lengths instead of limiting them for the computational issue. The phrases in pRNNs are composed and selected in a way that is jointly learned in the language modeling, therefore requiring no human-labeled data or external model such as word alignment. In previous RNN-based language modeling, the hidden state of RNN before the word to predict summarizes the history of all previous words. Similarly, in pRNNs, we use the all state of all parallel RNNs (with the same parameters) to capture the history of all sub-sequence of words that precede the word to predict, with the starting word shifting from the first word the one right before the word to predict.

This set of RNNs applied parallelly to different choices of word sequences are called RNN pyra-

mid. While most of those RNNs' status deal with incorrect word sequences: they could either start in the middle of a chunk, or in a place too early or too late for the prediction tasks, we left it to an attention mechanism to select and combine, therefore eliminate the need for external knowledge on chunking and composition. This mechanism will be trained jointly with the composition models in pRNNs in optimizing a designed objective function, e.g, perplexity or likelihood. With proper composition function in pRNNs, the RNN pyramid provides a "phrase forest", which could potentially contain a fairly deep nested structure in some of its members.

Our pRNN models have two merits:

- They represent all phrases in the same vector space in an explicit and unsupervised way. Which shows the potential to discover and utilize hidden structures of surface word sequences.

- They explore the possibility of network construction in another dimension: Parallel. Instead of stacking deeper and deeper layers of RNNs.

Experiments show that pRNNs are effective for language modeling (§4). Our model obtains significant better perplexities than state-of-the-art sequential Long-Short Term (LSTM) model on language modeling task, both on PTB and FBIS English data set. We also apply pRNNs to encode the source sentence of machine translation besides a conventional bi-direction encoder, which improves over the Moses (phrase-based statistical model) and a strong sequence-to-sequence baseline in the Chinese-English machine translation task.

## 2 Phrasal RNNs

We assume that, in the task of language model and machine translation, selecting the appropriate hidden structures for one sentence is highly related to the performance of the task.

Formally, a typical pRNN consist of three subnetworks: phrasal part $P$, attention part $A$ and sequential part $S$. Each sub-network plays its own role and collaborates with others.

$P$ (§2.1) constructs the neural structures (real-valued vectors) which corresponding to natural linguistic structures (phrases). It takes embed-

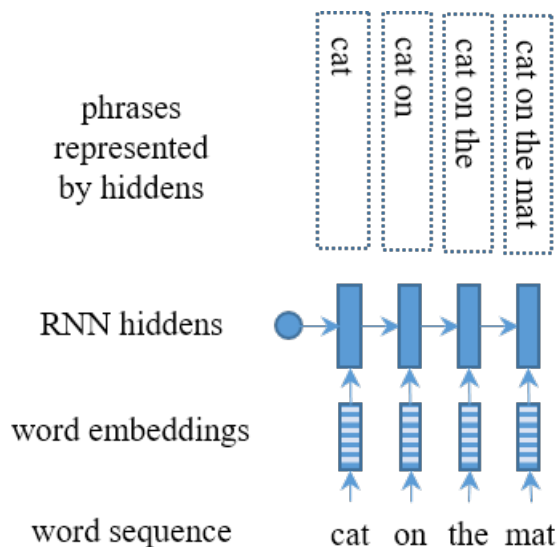

Figure 1: How original RNNs represent history with hidden status.

dings ($x_\star$) of all words in a candidate phrase, as input, then output one fix-length real-valued vector $p$ as the distributed representation (Hinton et al., 1986) of the phrase;

$$p = P(x_i^j) \qquad (1)$$
$$\text{Where } x_i^j = [x_i, \cdots, x_j] \qquad (2)$$

$A$ (§2.2) compares the candidate phrase regards to the current situation, give a probabilistic distribution over them, then provides the weighted sum of their representations. It takes previous calculated candidate set $\{p\}$ as input, then output a weighted sum of $\{p\}$ as $\hat{p}$ with the help of current hidden state ($h_t$) at time step $t$ given by ($S$).

$$\hat{p} = A(h, p_k, \cdots, p_l) \qquad (3)$$

$S$ (§2.2) combine the weighted sum of candidate phrases into original RNN, forces RNN taking the structural history information into consideration. It is similar to original RNN except for one point: when predict next hidden state $h_{t+1}$, besides $h_t$ and $x_t$, it takes $\hat{p}$ as input too.

$$h_t = S(h_{t-1}, x_t, \hat{p}_t) \qquad (4)$$

### 2.1 Represent Phrases

There are many types of neural networks which can transfer phrase into distributed representation. However, when we need to handle arbitrary length phrases, another way of saying, the entire history

of words, the choices are very limited to a few RNNs' variants.

Even when we choose RNNs to construct structure vectors, we are still facing a big problem. Because the hidden state $h_t$ are considered to encode the entire history information from the beginning of the sentence $x_1^t$, they can be utilized as representations of phrases begin at the sentence head $\{x_1^j | 1 \le j \le t\}$. But they do not provide the representations of phrases which do not begin with the first word of the sentence $\{x_i^j | 1 < i < j \le t\}$.

To represent all candidate phrases in a sentence with $n$ words, we build a RNN pyramid (RNNP (Fig. 3)), with $n$ horizontal parallel RNNs $\{RNN_n\}_{n=1}^N$. $RNN_n$ indicates that it begins at the $n$-th word of the sentence. With all $N(N+1)/2$ hidden status generated by RNN pyramid, we obtain distributed representation of all candidate phrases/structures of a sentence.

To keep consistent among these parallel RNNs in the pyramid and to limit the number of parameters for keeping the model simple, we let all parallel RNNs share the same network parameters $(W, U, b)$.

$$h_t^n = \sigma(W x_t + U h_{t-1}^n + b) \qquad (5)$$

Where $h_t^n$ of the $RNN_n$ indicates the hidden state of the $t$-th word in the sentence.

This method is kind of similar to the sharing parameters between filters of convolutional neural network (Cun et al., 1990), except for it working on the time axis, which recognizes the local invariant along each time steps.

With RNN pyramid built on a sentence, we can map all potential phrases with varying lengths into real-valued fix-length vectors. These vectors are representations of candidate structures we plan to compare at next stage.

## 2.2 Utilize Phrases

With the candidate structures represented by a fix-length vector (Fig. 2), we can easily apply attention mechanism on these vectors, and soft combine them to output a weighted sum as the best structure selected:

$$\hat{s}_t = \sum_{t,n} \alpha_{t,n} h_{t-1}^n \qquad (6)$$

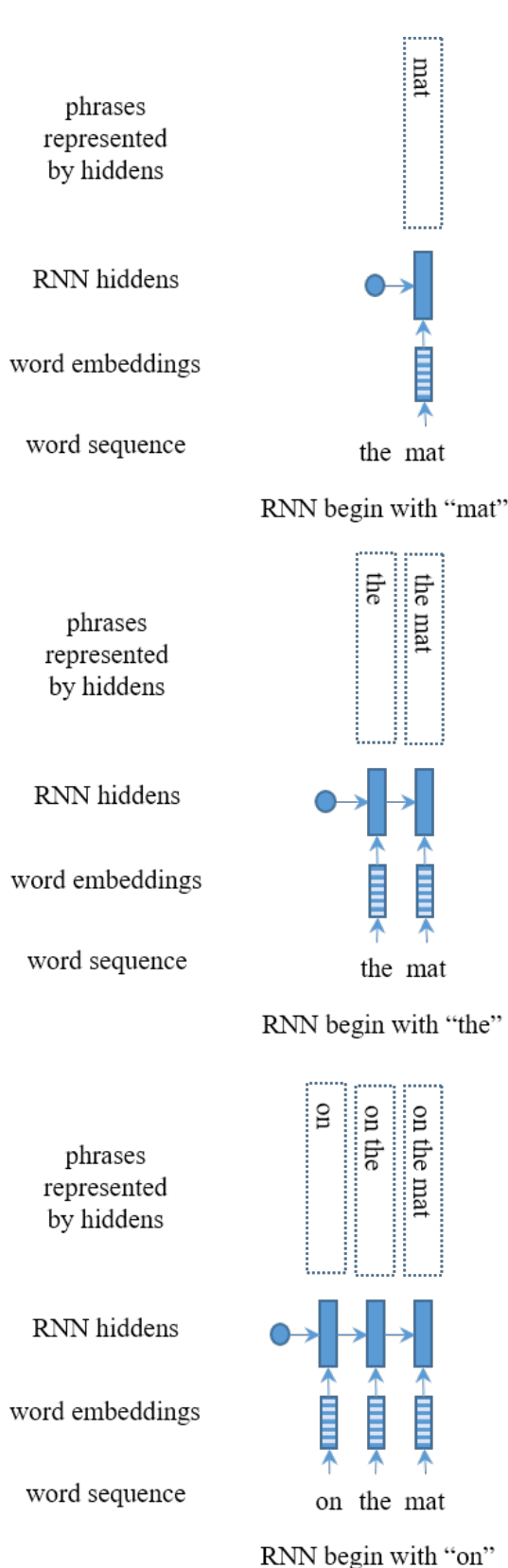

Figure 2: How shifted RNNs are able to represent phrases which do not start at the beginning of the sentence.

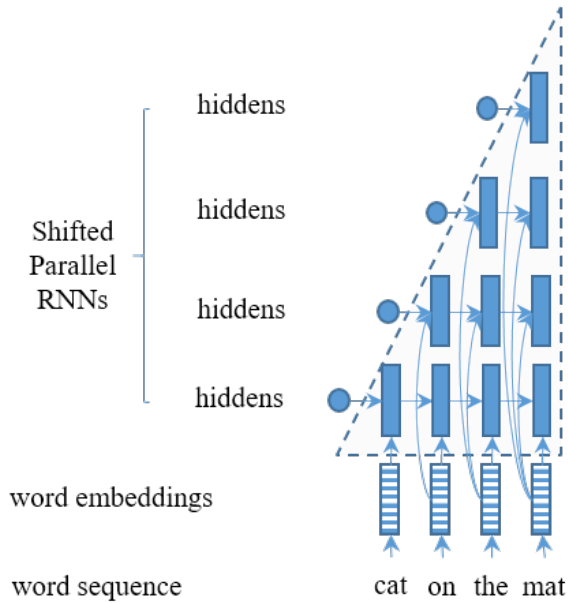

Figure 3: On a 4-word sentence, RNN pyramid (dashed line triangle) generated by 4 horizontal parallel RNNs, each begins at 1 of the 4 words in the sentence. Initial status are indicated by circles. Because hidden state is considered containing all history information. The set of all hidden status in the pyramid can be mapped one-to-one to the representation of all candidate phrases of the sentence.

Where the weight of hidden state (structure) $\alpha_{t,n}$ can be represented by the following form:

$$\alpha_{t,n} = \frac{exp(e_{t,n})}{\sum\limits_{t,n} exp(e_{t,n})} \qquad (7)$$

In which we define $e_{t,n}$ as:

$$e_{t,n} = a(h_k, h_{t,n}) \qquad (8)$$

Here we combine $a(h_k, h_{t,n})$ with one layer of feedforward neural network, where $h_k$ is the $k$-th word of sequential part $S$.

We adopt the attention mechanism from Bahdanau et. al. (2014). As we showed in Fig 4. We put $\hat{s}$ into $R$ part of the network, let the network to combine it with $h$ and $x$ to predict next hidden state. We also apply our model on machine translation task within successful Encoder-Decoder framework as in Fig 5.

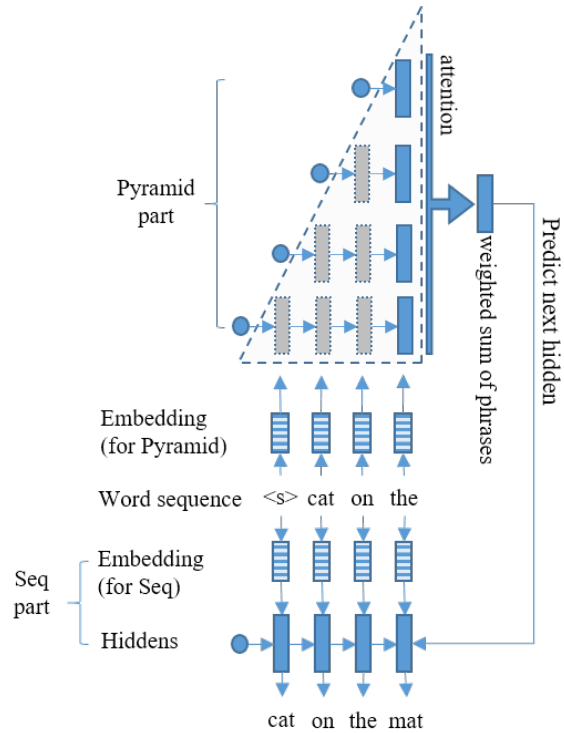

Figure 4: Combine best structure $\hat{s}$ given by attention part $A$ in each predicting step of sequential part $S$. To reduce calculation, we limit the candidate phrase set at each step to the newly generated ones (the blue rectangles with solid boundaries), which means to ignore structures generated at previous steps (grey ones with dashed boundaries), therefore reduce the scale of candidate phrases set from $O(N^2)$ to $O(N)$. The Pyramid and Seq part share the same embedding in experiments, we draw them separately in the diagram just for clearance.

## 3 LM Experiment

### 3.1 Data

To make experiment comparable with other methods, we apply our models on language model task, evaluate it in perplexity on the widely used English Penn Treebank (PTB) (Marcus et al., 1993), which pre-processing and splitting by Mikolov (2010). The data is utilized as following: sections (0-20) with 929k tokens are used for training, sections (21-22) with 73k tokens are held out as validation, and sections (23-24) with 82k tokens are used for testing. There are only top 10,000 high-frequency words are kept in the corpus. All rest low-frequency words are replaced with UNK tag. This version of data is widely used among the language modeling community. It is publicly avail-

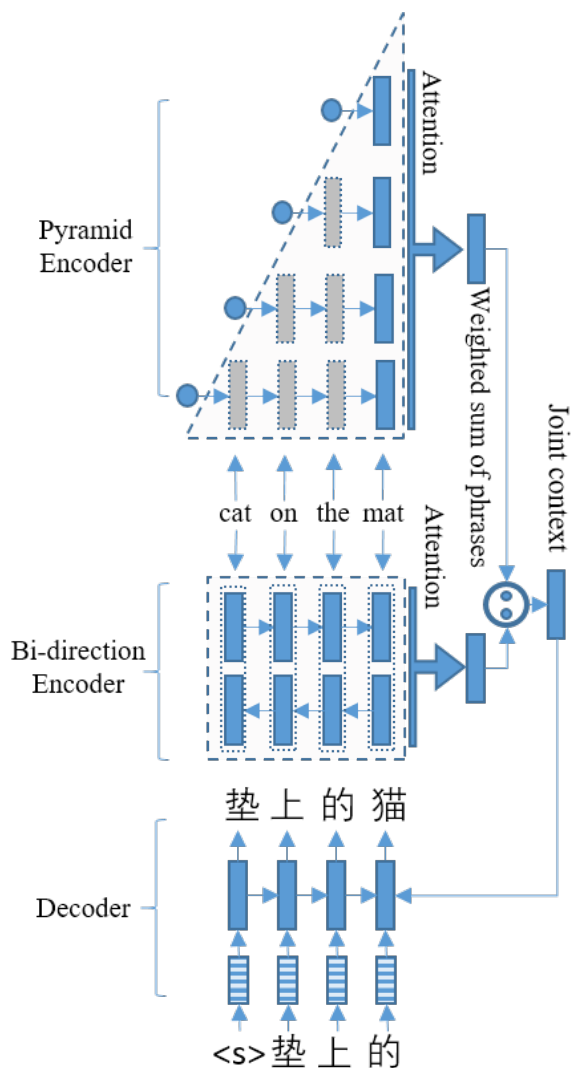

Figure 5: We use our pRNNs as the pyramid encoder to represent more structural information (all candidate phrases) of source sentences, just beside the original bi-direction encoder which only represents surface word sequence explictly. Then we join two context vectors from two encoders into a larger one. Then we allow the decoder to choose which portion of the larger context (which source words or candidate phrases) is more relevant to the next generated word of the target sentence. We adopt two settings of pyramid encoder, one takes only last status (the blue rectangles with solid boundaries) of each RNN as the input of attention part (src-pyr-last in Fig 4), the other takes all status (all rectangles including the grey ones with dashed boundaries) of all RNNs as the input of attention part (src-pyr-all in Fig 4).

able.[1]

Because the scale of PTB corpus is relatively small, we also train our model on larger FBIS English(LDC2003E14). Accordingly, we only keep top 40,000 high-frequency words in the corpus, replace rest low-frequency word with UNK tag. We use NIST MT06 as the validation set, NIST MT08 as the test set.

| | train | valid | test |
| | FBIS | MT06 | MT08 |
|---|---|---|---|
| Sequences | 219,280 | 6,560 | 5,424 |
| Tokens | 7,877,650 | 190,065 | 166,937 |
| Types | 49,210 | 8,476 | 9,576 |

Table 1: FBIS and NIST MT Corpus statistics.

### 3.2 Model Configuration

**Baseline** In this paper, we utilize state-of-the-art LSTM framework on language model proposed by Zaremba (2014) as the baseline model. Firstly, we stack 2 layers of LSTMs, to explore more abstracted patterns which are not supposed to be discovered by a single layer. Secondly, to increase the model's capability of noise tolerance and reduce the overfitting to training data, we introduce dropout (Hinton et al., 2012) before and after each recurrent layer. We choose dropout rate as 0.5 without tuning. For training, we used AdaDelta (Zeiler, 2012). To handle nan and inf which occasionally occur in gradient, we normalize the gradient of each batch to 1.0 and drop the parameter updates on such batches. To determine when to stop training, we set patience to 100. We set the dimension of both word embedding and hidden dimensions to 200. We initialized all parameters according to recommendations given in (Zaremba et al., 2014) and blocks(van Merriënboer et al., 2015). For all models, we used word embedding, hidden dimensions of 200 and 2-layer LSTMs. For both models, we choose dropout rate as 0.5. For training, we used AdaDelta and normalize the gradient to 1.0, set patience to 100. We initialized all parameters according to recommendations given in Zaremba et al. (2014).

**Phrasal RNN** We configure our model exactly as the baseline model, except adding an extra RNN

---

[1] http://www.fit.vutbr.cz/~imikolov/rnnlm/

pyramid layer above baseline's 2-layer LSTM. We also add dropout between 2nd LSTM layer and RNN pyramid layer. We set dimension of hidden state in RNN pyramid layer as 200 either. We utilize Gated Recurrent Unit (GRU) (Chung et al., 2014) to construct RNN pyramid layer. We also tried a simplified version of GRU to build the pyramid (pRNNv in table 2), which achieves the best result.

### 3.3 Results

| Model | Perplexity |
|---|---|
| 5-gram, KN5 | 141.2 |
| FFNN-LM | 140.2 |
| RNN | 124.7 |
| LSTM | 126 |
| genCNN | 116.4 |
| LSTM (baseline) | 106.9 |
| pRNN | **97.6** |
| pRNNv | **94.5** |

Table 2: Perplexity on PENN TREEBANK, where the top 5 rows of numbers are results reported in previous work, our baseline and new pRNN model are in last three rows.

| Model | Perplexity |
|---|---|
| 5-gram, KN5 | 278.6 |
| FFNN-LM(5-gram) | 248.3 |
| FFNN-LM(20-gram) | 228.2 |
| RNN | 223.4 |
| LSTM | 206.9 |
| genCNN | 181.2 |
| LSTM (baseline) | 171.8 |
| pRNN | **161.5** |

Table 3: Perplexity on FBIS data set, where the top 5 rows of numbers are results reported in previous work, our baseline and new pRNN model are in last two rows.

We report our perplexities result of language model in table (2 and 3). We calculate perplexity over a sequence $[w_1, \ldots, w_n]$ with

$$PPL = \exp\left(-\frac{log(Prob(w_1^n))}{n}\right) \quad (9)$$

(including the end of sentence (EOS) symbol). pRNN and its variant outperform over 10 points of

ppl over a strong baseline on both PTB and FBIS English data set.

## 4 MT Experiment

### 4.1 Data

We evaluate all three models, PBSMT, RNNsearch, pRNN on the same data set. We utilize 1.25M sentence pairs, which are extracted from LDC corpora as training data. There are 34.5 English words and 27.9M Chinese words in the training data. We select NIST 2002 (MT02) data set as our development set, the NIST 2003 (MT03), NIST 2004 (MT04), NIST 2005 (MT05), NIST 2006 (MT06) and NIST 2008 (MT08) as test sets. When training neural networks, we limit the size of vocabularies of both source and target side to the most frequent 16K words. All rest low-frequency words are replaced with UNK tag. Chinese vocabulary covers approximately 95.8% of the corpora. English vocabulary covers approximately 98.3% of the corpora.

### 4.2 Model Configuration

**Baseline** In this paper, we use an open-source implementation (Meng et al., 2015) of RNNsearch (Bahdanau et al., 2014) as baseline model. To increase the model's capability of noise tolerance and reduce the overfitting to training data, we introduce dropout (Hinton et al., 2012) before softmax layer, and set the dropout rate equal to 0.5. We choose dropout rate as 0.5 without tuning. For training, we used AdaDelta (Zeiler, 2012) and normalize the gradient to 1.0, . To handle nan and inf which occasionally occur in gradient, we normalize the gradient of each batch to 1.0 and drop the parameter updates on such batches. We set the dimension of both source and target word embedding as 620, and hidden dimensions to 1000. We initialized all parameters according to recommendations given in (Bahdanau et al., 2014). We also introduce the phrase-based model of Moses (Koehn et al., 2007) as a secondary baseline too.

**Phrase-based NMT** We configure our model exactly as the baseline model, except adding an extra RNN pyramid as a secondary source encoder. As the limitation on the memory of GPU, we keep only phrases ended at eos into consideration. Thus we name it src-pyr-last in table 4. We set the dimension of hidden dimensions inside pyramid as 1000 too. We utilize Gated Recur-

| Models | MT02 | MT03 | MT04 | MT05 | MT06 | MT08 | Test Avg. | Diff |
|---|---|---|---|---|---|---|---|---|
| moses | 33.41 | 31.61 | 33.48 | 30.75 | 31.07 | **23.37** | 30.056 | +0.754 |
| RNNsearch (groundhog) | 32.32 | 29.02 | 31.25 | 28.32 | 27.99 | 20.29 | 27.374 | -1.928 |
| RNNsearch (baseline) | 34.28 | 30.61 | 33.24 | 30.66 | 29.83 | 22.17 | 29.302 | +0.000 |
| pRNN (src-pyr-last) | 35.48 | 31.61 | 34.40 | **31.96** | **31.36** | 22.82 | 30.430 | +1.128 |
| pRNN (src-pyr-all) | **35.49** | **32.08** | **34.51** | 31.81 | 30.91 | 22.86 | **30.434** | +1.132 |

Table 4: BLEU score on 1.25M training corpus with 16k dictionary on both source and target side. The above two lines of the table are results of open-source machine translation systems. Bold numbers indicate the best results on the data set (column). pRNNs are better than original RNNsearch model baseline (in-house reimplemented). We find it is interesting that results of src-pyr-all are only slightly better than src-pyr-last, we guess this is due to the limited discriminative power of simple attention mechanism when meeting large number of complex candidates.

rent Unit (GRU) (Chung et al., 2014) to construct RNN pyramid layer.

For a fair comparison, we run baseline system (RNNsearch) many times (not epoch) and report only the best one. We only run PBNMT once.

### 4.3 Results

We report our BLEU result of three models in table (4). We use the case-insensitive 4-gram NIST BLEU (Papineni et al., 2002) score given by mte-val_v11.pl pRNNs outperforms both PBSMT and Encoder-Decoder model.

## 5 Related Work

### 5.1 Relation to Previous Attempts on Structural Information

In the last two decades, to achieve better performance, researchers incorporated more and more rich structural information into conventional model (Jelinek and Lafferty, 1991; Chelba, 1997; Chelba and Jelinek, 2000; Emami and Jelinek, 2005). This trend is more clear in statistical machine translation (SMT) community, from word-based SMT (Brown et al., 1993) to phrase-based SMT (Koehn et al., 2003), hierarchical phrase-based SMT (Chiang, 2005) and syntax-based SMT (Huang et al., 2006; Liu et al., 2006; Galley et al., 2006; Xie et al., 2011) (forest-based (Mi and Huang, 2008; Mi et al., 2008)). among them, phrase-based SMT (Zens et al., 2002; Koehn et al., 2003) was the most widely adopted translation model. The Same trend can be observed in neural network strand. There were many works which successfully modeled structure in neural network on parsing (Dyer et al., 2016; Emami and Jelinek, 2005; Henderson, 2004; Titov and Hender-

son, 2007; Buys and Blunsom, 2015) or language modeling tasks (Dyer et al., 2016; Chelba and Jelinek, 2000; Emami and Jelinek, 2005; Chelba, 1997).

In neural machine translation (NMT) area, the situation is more complex. In recent years, the most popular and success NMT model is Encoder-Decoder model (Bahdanau et al., 2014; Sutskever et al., 2014). Which has achieved competitive or better results in many translation tasks(Luong et al., 2015a,b). However, beyond sequential surface words, there are latent nest structures in natural language (Chomsky, 1957). One direction to explore is to introduce sub-word structures(Costa-Jussà and Fonollosa, 2016; Chung et al., 2016; Luong and Manning, 2016; Sennrich et al., 2015), such as characters. The most important reason to dig into sub-word is to handle out-of-vocabulary word problem. This problem is rooted in the limited size of vocabulary, which utilized by NMT mapping symbols to real-valued dense vector.

Another direction is to explore hyper-word structure. Authors of (Eriguchi et al., 2016) adopted parsing tree in encoder phase. However, this method depends heavily on human-labeled data, which is always expensive and limited in scale. Authors of (Stahlberg et al., 2016) introduce hierarchical phrase-based (HPB) model as the guider of search space. However, the HPB model and NMT model are trained separately and combined only when decoding. Compare this to the previous method, (Eriguchi et al., 2016) can be categorized into introducing external data, (Stahlberg et al., 2016) can be categorized into introducing external model. In an ideal situation, all external model can be replaced by a neural net-

work with equal ability.

## 5.2 Similarity to Deep Memory Network

Deep Memory Network is an effective implementation of the neural turing machine. When they use state machine such as GRU or LSTM to read from one memory and write to another, the memory IO addresses are either content-based (via attention mechanism) or location-based (actually sequentially cell-by-cell) (Meng et al., 2015). However, there are much more other methods in the location-based category.

**Cell-by-cell vs. Incremental** If we consider the hidden of one time step inside pyramid RNN as memory, we can name the operations as incremental read and write. The intuition behind incremental addressing is, when we read little, we know little, we only have the ability to write little. But when we read more, we know more, we are gone to have the ability to write more.

## 6 Discussion

Our experiments clearly show that the proposed pRNN model is quite effective in language modeling and machine translation. This is the because of:

- when model predicting, it is provided with all candidate phrases as structure information rather than just surface sequential words.

- utilizing attention mechanism to compare and combine to get the weighted sum which best fit for predicting next hidden state.

The most significant question that remains is how well the quality of forest generated as a by-product of pRNN, will it get a better result than other supervised parsing model trained on human label data.

## 7 Conclusion

We introduced phrasal recurrent neural network, an RNN model with all potential candidate phrases considered. Our model does not require any human labeled data to construct the structures. It outperforms the state-of-the-art LSTM language models. Our model does not require any external resources such as human labeled data or word align model to construct the phrases. It outperforms both state-of-the-art PBSMT and RNNsearch model.

We make two main contributions:

- Instead of packing all information in distributed representation and internal hidden status, which are computing-friendly, we try to represent natural structure in an explicit way, which are human-friendly.

- Instead of stacking deeper and deeper layers of RNNs, we explore the possibility of network construction in another dimension: making RNN sequences parallel.

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
