# Peer review of "Phrasal Recurrent Neural Network"

_ACL 2017 — decision unknown_

[Official Review · Reviewer 1 · rating 2 · confidence 4]
soundness 5 · originality 5 · clarity 2 · impact 3 · substance 4 · appropriateness 5 · meaningful comparison 3 · presentation format Poster

The paper describes an idea to learn phrasal representation and facilitate them
in RNN-based language models and neural machine translation

-Strengths:

The  idea to incorporate phrasal information into the task is interesting.

- Weaknesses:

- The description is hard to follow. Proof-reading by an English native speaker
would benefit the understanding
- The evaluation of the approach has several weaknesses

- General discussion

- In Equation 1 and 2 the authors mention a phrase representation give a
fix-length word embedding vector. But this is not used in the model. The
representation is generated based on an RNN. What the propose of this
description?
- Why are you using GRU for the Pyramid and LSTM for the sequential part? Is
the combination of two architectures a reason for your improvements?
- What is the simplified version of the GRU? Why is it performing better? How
is it performing on the large data set?
- What is the difference between RNNsearch (groundhog) and RNNsearch(baseline)
in Table 4?
-  What is the motivation for only using the ending phrases and e.g. not using
the starting phrases?
- Did you use only the pyramid encoder? How is it performing? That would be a
more fair comparison since it normally helps to make the model more complex.
- Why did you run RNNsearch several times, but PBNMT only once?

- Section 5.2: What is the intent of this section

[Official Review · Reviewer 2 · rating 2 · confidence 3]
soundness 5 · originality 5 · clarity 2 · impact 3 · substance 2 · appropriateness 5 · meaningful comparison 3 · presentation format Poster

This paper proposed a new phrasal RNN architecture for sequence to sequence
generation. They have evaluated their architecture based on (i) the language
modelling test evaluated on PTB and FBIS and (ii) Chinese-English machine
translation task on NIST MT02-08 evaluation sets. The phrasal RNN (pRNN)
architecture is achieved by generating subnetworks of phrases. 

Strengths
====

A new phrasal architecture. 

Weaknesses
====

**Technical**: 

It's unclear whether there is a limit set  on the phrase length of the pRNN.
Maybe I've missed this in the paper, if there is, please be more explicit about
it because it affects the model quite drastically if for every sentence the
largest phrase length is the sentence length. 

 - It's because if the largest phrase length is the sentence length, then model
can be simplified into a some sort of convolution RNN where the each state of
the RNN goes through some convolution layer before a final softmax and
attention. 

 - If there is a limit set on the phrase length of pRNN, then it makes the
system more tractable. But that would also mean that the phrases are determined
by token ngrams which produces a sliding window of the "pyramid encoders" for
each sentence where there are instance where the parameter for these phrases
will be set close to zero to disable the phrases and these phrases would be a
good intrinsic evaluation of the pRNN in addition to evaluating it purely on
perplexity and BLEU extrinsically. 

The usage of attention mechanism without some sort of pruning might be
problematic at the phrasal level. The author have opted for some sort of greedy
pruning as described in the caption of figure 4. But I support given a fixed
set of phrase pairs at train time, the attention mechanism at the phrasal level
can be pre-computed but at inference (apply the attention on new data at test
time), this might be kind of problematic when the architecture is scaled to a
larger dataset. 

**Empirical**: 

One issue with the language modelling experiment is the choice of evaluation
and train set. Possibly a dataset like common crawl or enwiki8 would be more
appropriate for language modelling experiments. 

The main issue of the paper is in the experiments and results reporting, it
needs quite a bit of reworking. 

 - The evaluation on PTB (table 2) isn't a fair one since the model was trained
on a larger corpus (FBIS) and then tested on PTB. The fact that the previous
study reported a 126 perplexity baseline using LSTM and the LSTM's perplexity
of 106.9 provided by the author showed that the FBIS gives an advantage to
computing the language model's perplexity when tested on PTB.

 - Also, regarding section 3.3, please cite appropriate publications the
"previous work" presented in the tables. And are the previous work using the
same training set? 

- Additionally, why isn't the the GRU version of pRNNv reported in the FBIS
evaluation in Table 3?

The result section cannot be simply presenting a table without explanation:

 - Still on the result sections, although it's clear that BLEU and perplexity
are objective automatic measure to evaluate the new architecture. It's not
really okay to put up the tables and show the perplexity and BLEU scores
without some explanation. E.g. in Table 2, it's necessary to explain why the
LSTM's perplexity from previous work is higher than the author's
implementation. Same in Table 3. 

The result presented in Table 4 don't match the description in Section 4.3:

 - It's not true that the pRNN outperforms both PBSMT and Enc-Dec model. The
authors should make it clear that on different evaluation sets, the scores
differs. And it's the averaged test scores that pRNN performs better

- Please also make it clear whether the "Test Avg." is a micro-average (all
testsets are concatenated and evaluated as one set) or macro-average (average
taken across the scores of individual test sets) score. 

For table 4, please also include the significance of the BLEU improvement made
by the pRNN with respect to the the baseline, see
https://github.com/jhclark/multeval

General Discussion
====

As the main contribution of this work is on the phrasal effect of the new RNN
architecture, it's rather important to show that the phrases are more coherent
than the vanilla LSTM / RNN model. Thus the BLEU evaluation is insufficient. A
closer look at evaluating the phrases in a subset of the evaluation set would
be necessary to support the claims. 

Does the baseline system (groundhog) contains the attention mechanism? 

 - If so, please be more specific in describing it in section 4.2 and Table 4. 

 - If not, please remove the attention layer after the encoder in figure 5.
Also, the lack of attention mechanism provides a disadvantage to the baseline
enc-dec system and it's unclear whether the pRNN can outperform or be an
additive feature to the enc-dec system with an attention mechanism. The unfair
disadvantage is even more prevalent when the pRNN uses multiple phrasal
attention layers within a single sentence while a simple enc-dec system without
attention is used as a benchmark =(

Question: Wouldn't a simpler way to get phrasal RNN is to put the "pyramid"
RNNs of a phrase into some soft of a average pooling layer?

Minor Issues 
====

Figure 2 is a little redundant, I think figure 1 is enough to compare it
against the pRNN (figure3 and 4).

Also, possibly figure 3 can be combined into the pyramid part of figure 4. And
more space can be freed up to further explain the results section. 

Please don't abuse figure/table captions, whenever possible, please try to keep
the description of the tables and figures in-text.  

**Please put the verbose caption description in the main text for Figure 3, 4,
5 and Table 4**

Spacing in between some of the equations can also be reduced (e.g. in latex use
\vspace{-5mm} )